# Fabrication of Electrospun Polymer Nanofibers with Diverse Morphologies

**DOI:** 10.3390/molecules24050834

**Published:** 2019-02-26

**Authors:** Chenyu Wang, Jun Wang, Liangdan Zeng, Ziwen Qiao, Xiaochen Liu, He Liu, Jin Zhang, Jianxun Ding

**Affiliations:** 1Department of Orthopedics, Hallym University, 1 Hallymdaehak-gil, Chuncheon, Gangwon-do 200-702, Korea; cathywang0111@hotmail.com; 2College of Chemistry, Fuzhou University, Fuzhou 350116, China; junwang0591@sina.com (J.W.); lllxiaochen@163.com (X.L.); 3College of Chemical Engineering, Fuzhou University, Fuzhou 350108, China; LiangdanZeng@163.com (L.Z.); ziwenqiao@126.com (Z.Q.); 4Key Laboratory of Polymer Ecomaterials, Changchun Institute of Applied Chemistry, Chinese Academy of Sciences, Changchun 130022, China; heliu@ciac.ac.cn (H.L.); jxding@ciac.ac.cn (J.D.)

**Keywords:** polymer nanofiber, fabrication, electrospinning, morphology, application

## Abstract

Fiber structures with nanoscale diameters offer many fascinating features, such as excellent mechanical properties and high specific surface areas, making them attractive for many applications. Among a variety of technologies for preparing nanofibers, electrospinning is rapidly evolving into a simple process, which is capable of forming diverse morphologies due to its flexibility, functionality, and simplicity. In such review, more emphasis is put on the construction of polymer nanofiber structures and their potential applications. Other issues of electrospinning device, mechanism, and prospects, are also discussed. Specifically, by carefully regulating the operating condition, modifying needle device, optimizing properties of the polymer solutions, some unique structures of core–shell, side-by-side, multilayer, hollow interior, and high porosity can be obtained. Taken together, these well-organized polymer nanofibers can be of great interest in biomedicine, nutrition, bioengineering, pharmaceutics, and healthcare applications.

## 1. Introduction

Micro/nanofiber mats have been a subject of intensive research due to their high specific surface area and interconnected porous structure [1]. These unique features, in addition to the intrinsic functionalities of polymers, impart nanofiber mats with many desirable properties for interesting applications in a multitude of fields [2,3,4]. Several methods have already been applied for generating micro/nanofiber mats, such as phase separation [5,6], template synthesis [7], melt-blown [8,9], self-assembly [10,11], three-dimensional (3D) printing [12,13], and electrospinning [14,15,16]. The advantages and disadvantages of each of these techniques are summarized in Table 1. More importantly, electrospinning allows the fiber diameter to be adjusted from nanometers to microns [17]. Various macromolecules have been successfully electrospun into the ultrafine fibers. In terms of process flexibility, electrospinning is capable of producing continuous nanofibers from a wide variety of materials.

The concept of electrospinning was first proposed by Formhals when he obtained a patent in 1934 to draw polymers into fine filaments with the application of an electrical charge [18]. Up until the mid-1990s and with increasing interest in the field of nanoscience and nanotechnology, several research groups started to fabricate electrospun fibers from a broad range of polymers [19,20]. The shape and diameter of the obtained polymer structure are not only affected by the molecular weight, viscosity, electrical conductivity, surface tension [21], but also the voltage applied during electrospinning, the distance from the tip to the collector [22], and the feeding speed [23]. The ideal state of the electrospun polymer was obtained by regulating these parameters.

With the ability to fabricate nanostructures from a wide variety of raw materials, including natural and synthetic polymer composites (both organic and inorganic) [26], electrospinning is attracting more and more scientists for the highly-efficient preparation of various nanostructures. Electrospun nanofibers have broad application prospects in various fields, such as tissue templates, drug delivery, pharmaceutical ingredients, medical prostheses, artificial organs, wound dressings, bone tissue engineering, filtration formation, and sensing [27]. Specifically, electrospun collagen fibers can improve the interaction between cells and scaffolds, ultimately resulting in enhanced attachment, proliferation, and differentiation of cells [28]. Unfortunately, even though with the widespread use, the understanding of the electrospinning is still minimal. In the review, we first introduced the basic knowledge of electrostatic spinning, including device, composition, mechanism, and methods comprehensively. Then, considering the relationship among structure, property, and application, some unique structures and morphologies of electrospun fibers reported in current studies have been summarized, as shown in Figure 1.

## 2. Fabrication of Electrospun Polymer Nanofibers

### 2.1. The Basic Setup and Composition for Electrospinning

The primary electrospinning apparatus consists of four major components: a high-voltage source which creates an electrical field between a positively-charged syringe needle and a grounded collector, a metallic needle where the charged solution is forced to stretch under the electrostatic forces, a syringe pump, and a grounded target to deposit the resultant fibers [29]. Electrical wires connect the high power supply to the metallic needle, and there maintains a relatively short distance between the syringe tube and target. Figure 2A shows a schematic illustration of the typical electrospinning set-up. As the solvent evaporates while electrospinning, the jet will be elongated by electrostatic repulsion. This is followed by the thinning process which leads to the formation of a uniform fiber within micro- to nanoscale, which can be collected in various orientations to create some specialized structures with different composition and mechanical properties. To date, various targets have been employed to collect fibers in the course of electrospinning, mainly including aluminum foil [30,31], copper plates [32], and rotating drums [33,34]. With the application of this relatively straightforward technique, more than 40 different types of natural and synthetic organic polymers have already been successfully electrospun to fibers with diameters ranging from tens of nanometers to a few micrometers [35,36,37].

The electrospun micro/nanofibers produced nowadays are not only based on polymers but also on ceramics, metals, metal oxides, organic, and inorganic composite systems. These electrospun membranes composed of copolymers, blends, or organic fillers always exhibit enhanced mechanical behavior, barrier properties, and thermal stability [40,41,42]. A comprehensive summary of electrospun polymers, the solvents used, the polymer concentrations in different solvents, and the prospective applications of corresponding nanofibers are listed in Table 2.

Drug-containing electrospun fiber mats have gained widespread interest in various biomedical applications, including wound dressing, tissue remolding, and prevention of the anaerobic bacteria colonization, and so forth [43]. Compared with the low drug delivery efficiency of microspheres, hydrogels, and micelles systems, fibrous carriers are more promising because of their relative ease of use and flexible adaptability. However, there are still many problems for researchers to solve, especially for the burst release of drugs from the sample surface during the first 10–12 h. In order to avoid this phenomenon, researchers are investigating why the burst release occurs and how to achieve a constant drug release profile.

Different proteins and genes have been loaded into the electrospun scaffolds mainly by the following four ways (Figure 3). The easiest way is to dip the scaffolds into an aqueous phase containing biomolecules, where the loaded biomolecules can attach to the scaffolds via electrostatic forces [75]. In the second situation, biomolecules are mixed with the polymer solution. Such blend electrospinning localizes biomolecules within the fibers of the scaffolds rather than the superficially physical adsorption, guaranteeing a more sustained release profile [76]. Coaxial electrospinning or emulsion electrospinning, are other promising ways in producing the core–shell fibers to preserve the activities of proteins [76]. More importantly, the core/shell nanofibers have been reported to provide typical biphasic drug release profiles consisting of an immediate and sustained release. The amount of drug released in the first phase is tailored by adjusting the shell flowing rate, and the remaining drug released in the second phase is controlled by a typical diffusion mechanism [77]. The fourth fabrication method is to immobilize the biomolecules onto the fiber surface via a chemical bond, through which the release rate of the immobilized biomolecules can be controlled by the external enzymes [78].

### 2.2. Mechanism of Electrospinning

The electrospinning technique involves the application of a high-voltage electrostatic field to overcome the surface tension forces of the droplet. In detail, the electrostatic repulsion and Coulomb forces provided by the external field are the main driving forces to induce the eruption of a charged polymer solution through the tip of the spinneret [80]. Through increasing the strength of the electrostatic field, the surface liquid gets electrically-charged, and the shear stresses are produced by the repulsion between these charges. As these forces are opposite to the surface tension direction, the initial hemispherical drop elongates and transforms gradually. It is noteworthy that as the electric field strength increases to a critical point, a diaphanous and conical protrusion called the Taylor cone is formed (as shown in Figure 3A) [81,82].

Afterward, a nearly straight jet develops from the cone. Simultaneously as the jet travels toward the collector, the polymer solution undergoes stretching and whipping while the solvent evaporates. Although the jet is stable near the tip of the spinneret, it soon enters an unstable bending stage with further stretching of the solution jet under the electrostatic forces in the solution as the solvent evaporates. In addition, the perimeter of each turn of the coils grows monotonically (as shown in Figure 2B) [39,83]. In the end, the solid electrospun fibrous matrices are stacked on the grounded collector.

Morphologies of the electrospun fibers could be affected by the following parameters: (i) processing variables (e.g., electric potential [84,85], flow rate [86], collector set-up [87], and capillary tip-to-collector distance [88]); (ii) system characterizations (e.g., molecular weight of the polymer [89], viscosity [90,91], conductivity [92], and surface tension of the polymer solution [93]); (iii) ambient conditions (e.g., temperature [94] and humidity [95]), all of these have been widely studied in several recent publications [16,96]. It was found that the strength of the applied electric field would influence the shape and diameter of the electrospun fibers [97]. Increasing the applied voltage always leads to a larger fiber diameter; however, a field strength that is too strong will cause bead defects in the electrospun fibers. In general, fibers become thicker as the concentration of the solution increases, with the exception of solutions that are too dilute or too concentrated, in which the fibers will collapse into the droplets or cannot be extruded due to its high polymer entanglements [98].

The type of collector utilized can also influence the structural morphology of the electrospun fibers. Copper mesh, aluminum foil, water, and paper are typically employed to collect the electrospun polymer fibers. By comparison, a non-conductive collector always creates a highly porous structure with circular pores on the fiber surfaces. This is because the non-conductive collector cannot dissipate the charge among the fibers leading to a decrease in packing density and an increase in porosity [99]. Moreover, the distance between the tip of the syringe and the collector is another critical factor influencing the electrospinning process. It was reported that a distance that is too small or too large would result in the generation of polymer beads [100]. Therefore, by varying these parameters above and determining the appropriate values, we can generate electrospun micro/nanofibers with desirable morphologies, diameters, and minimal bead-on-string formation.

### 2.3. Modification of the Electrospinning Setup

To overcome various limitations of the basic electrospinning set-up and to further tailor the performance of resultant fibers, researchers have devised numerous methods to modify the set-up for electrospinning, in particular, the spinneret and collection devices [101,102,103]. A schematic diagram of the modified set-up is shown in Figure 4. To the best of our knowledge, one of the earliest electrospinning inventions were proposed by Fennessey and Farris, who claimed that it was possible to macroscopically produce one-dimensional (1D)- as well as 3D-aligned fibers by using a rotating mandrel collector [104]. Several other collectors have been proposed to acquire well-aligned fibers such as incorporating a copper wire drum [105], conductive plates containing an insulating gap [80], or a scanning tip [106] to the fiber collecting system. The rotating device can mechanically stretch the fibers toward the roll-up direction, thus helping the fibers to align along the periphery of the mandrel. Matthews et al. investigated the effect of mandrel rotation speed on the diameter, orientation degree, and material properties of the fibers [107]. It was noted that alignment of the fibers was improved when the rotation speed was increased from 800 rpm to 2000 rpm. At a sufficiently high rotating speed, the highly aligned fibers would induce the response of gene expression and cell interaction along the fiber orientation [82,108].

In addition to the aligned two-dimensional (2D) fiber mesh, a 3D conduit with a degree of orientation can also be fabricated by depositing fibers over a rotating rod (d < 5 mm). This could be applied for the construction of vascular grafts [109,110]. Similarly, the collector configurations for oriented fibers can also be modified by placing a group of counter electrodes in a specific position. Thontree et al. proposed to control the molecular orientation of electrospun fibers by using two parallel counter electrodes separated by a short air gap. They were able to successfully generate extended chain crystals (ECC) of polyoxymethylene (POM) [111]. These electric field concentrators exert a tensional electrostatic force on the electrospun jet, which results in the stretching effect of the jet to the linear array between two given edges. In summary, both the rotating device and the counter electrodes significantly enhance the alignment of the electrospun fibers. The acquired anisotropy degree of an electrospun fibrous mat can individually affect the mechanical properties of fibers, cell adhesion, proliferation, and alignment.

Multicomponent fibers are important for applications in the areas of nanosprings [112], super-hydrophobic surfaces [113,114], sensors [115,116], and drug delivery [117,118]. The core–shell structures, which include a core and any number of shells, are the most prominent multicomponent fibers [13,119]. Accordingly, a modified spinneret with coaxial capillaries has been developed for building these materials [120,121]. The drugs can be introduced into the core or the sheath of the nanofiber to meet the emerging needs of multifunctional devices. It should be noted that encapsulation of non-spinnable core material inside a spinnable shell material is one of the advantages of this particular structure. Additionally, the coaxial electrospinning could provide excellent control over the structure and morphology of the resultant fibers. By choosing various types of polymer fluids and precisely controlling the processing parameters of the system, different physical/biological properties could be imparted to the nanotubes or core–shell nanofibers [122]. As proved by Sebastian et al., a more uniform deposition of titanium dioxide (TiO_2_) nanoparticles was shown in the collecting substrate of electrically conductive polyaniline (PANI)-poly(ethylene oxide) (PEO) nanofibrous membrane, compared with the fiber mat placed on aluminum foil. Moreover, the catalytic activity of the blended membrane was improved with an increase in the concentration of TiO_2_ [123]. This revolutionary technique was promising, not only for the encapsulation of biomacromolecules or nanoparticles but also for the modification of fiber surfaces [124].

Jiang et al. designed another notable variation of this configuration and described a novel multi-fluidic compound-jet electrospinning technique where three metallic inner capillaries were arranged as an equilateral triangle [125]. By separately feeding different viscous liquids into the three inner capillaries and an outer syringe at an appropriate flow rate, the multichannel tubes have received great interests as of late due to unique combinations of various functionalities. Accordingly, these novel electrospun membranes showed broad potential in biomimetic materials [83,126], high efficient catalysts [127], and multi-component drug delivery applications [128].

It has been observed that controlling the deposited density and area of electrospun fibers will widen the application spectrum of these mats. Natural bone is a complex biomineralized system with excellent mechanical stability, highly-densified mineralization matrix, as well as a complex hierarchical structure. Given these characteristics, design of the scaffolds should mimic the morphology as well as functional structure of the extracellular matrix (ECM). Layer-by-layer (LbL) electrospinning is a good candidate for acquiring hierarchical-structured scaffolds because each polymer solution will be electrospun to form its layer, and subsequently deposited on the same target metallic collector in the form of a nonwoven fabric [129]. A bilayered hybrid scaffold comprised of unique traits from PLGA microfibers and naturally derived acellular matrix was fabricated [130]. This scaffold could support tissue regeneration and held great potential for hollow engineering organs.

Mikos et al. have also developed an on-site LbL technique to construct a multilayered 3D scaffolds consisting of alternating micro- and nanofiber layers [131]. In this approach, the effect of fiber layer thickness on bone marrow stromal cells (BMSCs) attachment, spreading, and migration was intensely discussed. Additionally, upon utilizing the LbL electrospinning technique, Layman et al. obtained a functional biohemostat, a form of a super-absorbent polymer membrane for treating high-pressure bleeding [132].

One novel configuration composed of a multi-needle system and an electrically-charged cylindrical electrode is designed to increase the deposition density of the fiber mat [133,134]. The setup consistently encloses the needle system inside an iron ring, which is helpful to improve the concentration of electric field lines, reduce the deposited area, and considerably increase the throughput of electrospun fibers. This innovative approach has been employed to fabricate a vascular scaffold that dual-loaded vascular endothelial growth factor and platelet-derived growth factor. This multilayered fibrous scaffold has been demonstrated to benefit blood vessel reconstruction, facilitate endothelialization by the dual release of the growth factors, and inhibit hyperproliferation of vascular smooth muscle cells [135,136,137].

Twisted nanofibers have been extensively exploited in the field of tissue engineering nowadays because they can readily emulate several natural materials, such as collagen fibrils and double-stranded DNA for use in clinical applications [138,139]. A number of electrospinning setup modifications are being analyzed to configure such twisted nanofibrous yarn. One of these modifications is employing a liquid container to collect the electrospinning micro/nanofibers. In most cases, a solid collector is generally-applied to deposit the electrospun fibers. In the method proposed by Smit et al., a mesh consisting of random electrospun fibers was deposited on a liquid surface for the first time [140]. This was done to neutralize the free charges available on the surface of the fibers. Then, these neutralized fibers were subsequently drawn to a rotating mandrel to obtain the fiber yarn.

In addition, by studying the influence of liquid types on the properties of the yarn, it was found that a liquid medium with high surface tension, such as water, was significantly more preferred to collect the fibers [141]. Another popular method to acquire twisted nanofibers is using an auxiliary electrode [142,143]. When the electrical charge was applied to each side of the auxiliary electrodes, the electric field rotated around the auxiliary electrode in a sequence and finally resulted in a 360° rotation of electrospun jet. It was also reported that regulating the rotation time, as well as the amplitude of the electric field on the auxiliary electrode is able to control the twist length of the yarn [144]. In conclusion, twisted nanofibers fabricated directly using a modified electrospinning system have great potential to be utilized in applications as artificial muscles and actuators.

### 2.4. Electrospinning Methods

In addition to the modification of the spinneret and the collection devices, the choice of the electrospun matrix is another method that can be modified for electrospinning. Typically, there are two available methods for electrospun polymers. The first method involves dissolving the polymer in a suitable solvent and electrospinning it. In the second method, the polymer can be directly electrospun from a melt. Both of these methods require the polymer to be in a liquid state, either by heating or dissolving the polymer. Afterward, the liquid is electrically charged and dispensed in an orderly format to create different types of microstructures [148]. Each method has its specific advantages and disadvantages. For instance, solution-spinning results in a greater range of fiber size from nano- to micrometers, and it can be stably processed at room temperature. In accordance with the melt electrospinning method, Dalton et al. used polymer melts to deposit fibers for tissue engineering which proved to be ideal with regard to the highly reproducible production and low manufacturing costs. This is because the melt electrospinning method eliminates any chances of introducing harsh organic solvents [149].

Moreover, annealing can further modify the properties of the electrospun membrane, including thermal behavior and mechanical properties [150]. However, the drawback of melt electrospinning is the non-uniformity of the fiber diameters due to drawing instabilities. Additionally, it is very difficult to fabricate melt-spun fibers with nanometer diameters, and the polymers must be kept at elevated temperatures above the melting point [151,152].

The melt electrospinning membrane has been widely-used as scaffolds for different engineering types of tissue. Such scaffolds have been reported to combine with cells and other biological components for replicating the tissues found in nerves, muscles, cartilage, bone, skin, and tendons [153]. For example, melt electrospun poly(hydroxymethyl glycolide-*co*-ε-caprolactone) (pHMGCL) fiber scaffolds were demonstrated to improve the cellular response to the mechanical anisotropy vastly. The cardiac progenitor cells were able to align more efficiently along the preferential direction of the melt electrospun pHMGCL fibers compared to commonly-used electrospun scaffolds, hereby potentially enhancing their therapeutic potential in cardiac tissue engineering [148].

Loessner et al. built a novel class of melt electrospinning devices for fabricating scaffolds with different surface characteristics, which could support the growth of various cell types to deposit their own ECM and mimic the natural microenvironment in vitro [154]. As shown in Figure 5, melt electrospinning scaffolds were able to develop an endosteal bone-like tissue to promote the growth of human hematopoietic stem cells. More importantly, a significantly-enhanced deposition of endosteal proteins and osteogenic markers could be observed when the tissue-engineering scaffold was combined with calcium phosphate coating under osteogenic conditions.

## 3. Diverse Morphologies of Electrospun Polymer Nanofibers

Recently, researchers have begun to look into various applications of electrospun fibers because they possess several prominent advantages. Firstly, compared to other approaches of generating fibers such as mechanical drawing [155,156], phase separation [157], and self-assembly techniques [158], electrospinning is better suited in terms of its flexibility, simplicity, and ease of high-volume production [159]. Through the application of an external electric field, the uniaxial elongation derived from the electrostatic repulsions is able to generate continuous fibers on a large scale. Secondly, in terms of adaptability, the versatility of the electrospinning technique has allowed for producing a vast range of materials, including polymers [160], composites [161,162], semiconductors [163], and ceramics [164,165]. Additionally, previous studies have shown that electrospun matrices comprised of nanofibers have the extremely high specific surface area to interact with cells [166,167], making them ideal for cell attachment and proliferation. Accordingly, an attractive feature of electrospinning is its capacity to adjust fiber size in the nanometer and submicron range, which closely resembles the size of extracellular structures. Thirdly, the distinct advantage of electrospun matrices appears to be the exceptionally high surface-to-volume ratio [168].

Interestingly, by regulating the solution and electrospinning parameters, porous fibers with the increased surface area could be attained [169,170]. In addition, depending on the entanglement of these micro/nanofibers, the electrospun membranes possess a highly-porous 3D network with excellent pore interconnection [171,172]. These mats can mimic diversified ECM in terms of texture and compositions (dependent on the choice of materials taken) making them excellent candidates for use in tissue engineering. Last but not the least, the electrospun polymer chains are always aligned along the fiber axis because they experience a rapid stretching force. As a result, some performance differences are presented among the chain orientation of the electrospun fibers, especially the thermal behavior and physical-mechanical properties. According to Pedicini et al., in contrast to the resultant products obtained by solution casting or other conventional processes, electrospun PU fibers exhibit a distinctly-different stress-strain response curve in the uniaxial tensile test [173]. These aforementioned unique characteristics of electrospun fibers impart the matrices with many desirable properties. However, for their potential to be fully-realized and to achieve superior performances, the further design of various fibrous assemblies and morphological structures are necessary.

### 3.1. Core/Shell Structures

The main advantages of nanofibers with core–shell structures are successfully deferring the initial burst release and protecting the bioactivities of drugs [120]. In detail, the programmed release could be realized by an impermeable shell that provides temporary protection of the drugs within the core. With the use of a conventional electrospinning setup, it is possible to observe the formation of core–shell structures, especially for a solution containing immiscible polymers that will phase separate as the solvent is evaporated [174]. Recent advancements imply that the core–shell nanofibers could be acquired by coaxial electrospinning and emulsion electrospinning.

For coaxial electrospinning, two different polymer solutions are pumped through a spinneret comprised of coaxial capillaries [143,175,176,177]. This setup allows for different solutions to be utilized in each nozzle as well as separate flow rate control. In order to get well-defined core–shell structures, two crucial aspects should be considered. One is the miscibility of the polymers and the solvents that appear in the core and shell solution, which will impact the integrity of the final core–shell architecture. Another critical variable is the solution flow rate of the shell and core polymers as both of them can be controlled to determine the shell thickness and core diameter [178]. Using this particular configuration of electrospinning setup, a smaller fiber can permanently be encapsulated by a larger fiber leading to the core–shell morphology. This technique proves to be versatile not only for modifying the surface properties of electrospun fibers but also for the encapsulation of any drugs or biomacromolecules.

Lee et al. reviewed the core–shell nanofibers by a surface-modification technique based on oxygen-plasma treatment and coaxial electrospinning using PCL as the core and collagen as the surface shell [179]. The presence of a collagen shell was conducive to facilitate the migration of neural cell inside the scaffolds. Therefore, the coaxial-electrospinning nanofibers were demonstrated to possess higher cell proliferation efficiency in comparison to the normal nanofibers as well as the solution-coating nanofibers.

In addition to the function of surface modification, the core–shell bicomponent nanofibers can also effectively control the release kinetics. BMP-2 and DEX were encapsulated into PLLACL/collagen nanofibers by the coaxial electrospinning method. From the release profiles of the two growth factors, it can be seen that the core–shell nanofibers showed more controlled release compared to the blended electrospun fibers. Furthermore, this controlled behavior of BMP-2 and DEX-induced hMSCs to differentiate into osteogenic cells which were better for bone tissue engineering [180] (Figure 6). Additionally, for antibacterial applications, Zheng et al. successfully fabricated the drug-loaded electrospun non-wovens by coaxial electrospinning approach. In this case, the loaded amoxicillin within the nano-HAP/PLGA hybrid nanofibers exhibited a sustained release profile and non-compromised activity to reduce the growth of a model bacterium [181].

Generally, the core aqueous solutions loaded with enzymes or growth factors are not able to electrospun by themselves because the viscosity and concentration of the liquid are so low that it is impossible to stretch the core into a continuous thread within the sheath [182]. To solve this problem, the core emulsion was prepared by mixing an aqueous phase with a polymer solution. Another important feature of coaxial electrospinning is that it can be successfully utilized to acquire various nanofibers due to its core–shell structure, where the composition of the core can be varied in a broad range. For instance, Yu et al. encapsulated ketoprofen in the core of zein nanofibers and these core–shell nanofibers presented a linear drug release over a period of 16 h via gradual diffusion [183].

Furthermore, in order to accelerate the proliferation of vascular endothelial cells and vascular smooth muscle cells, electrospun membranes loaded with vascular endothelial and platelet-derived growth factors were developed as the inner layer by employing a modified coaxial electrospinning technique [184]. Small molecules like ketoprofen, macromolecules such as growth factors, mixtures of PLGA/BSA [185], and even cells [184] all had the ability to sufficiently maintain the biological activity for a long duration of time upon coaxial electrospinning.

Recently, a novel approach named “emulsion electrospinning” has attracted interests for fabricating core–shell fibers. It is remarkable that, unlike the special apparatus of coaxial electrospinning, the necessary equipment required for emulsion electrospinning is only a single needle. Additionally, the emulsion electrospinning was demonstrated to perform better than the conventional coaxial electrospinning, with respect to controlled drug delivery [185]. As described in our previous paper [144], the emulsions used for electrospinning usually contain an oil phase of the polymer solution and an aqueous phase, in which drugs or biomacromolecules are dissolved. Under electric forces, a uniform core–shell structure formed due to the stretching and coalition of the emulsion. Such composite fibers typically have a hydrophobic polymer sheath and a hydrophilic core, which would be fabricated into bioactive tissue-engineering scaffolds [186]. Confocal laser scanning microscope (CLSM) images of the resultant nanofibers which consist of green core and colorless sheath show that the boundary between them is quite sharp. It was also found that the volume ratio of the core to the shell could be varied by adjusting the emulsion concentration and emulsification parameters [187].

A mechanism involving “evaporation and stretching induced de-emulsification” was proposed to explain the transformation from emulsion droplets to core–shell fibers. The emulsion droplets were stretched into an elliptical shape along the fiber direction, which might be caused by the relatively-rapid elongation and the quick evaporation of the solvents during the electrospinning. Additionally, the viscosity gradient that existed between the elliptical droplets and their matrix resulted in the inward movement and merging of the emulsion droplets [188].

### 3.2. Hollow Interiors

Nanostructures with hollow interiors have attracted increasing attention due to their abundant applications, including drug release, nanofluidics, gas storage, sensing, energy conversion, and environmental protection [189,190,191,192]. A variety of methods have been proposed to generate these hollow nanotubes. For instance, self-assembly of the organic building blocks was once utilized to fabricate the tubular nanostructures. However, there were some significant limitations in the large-scale synthesis and strict structure control for this approach [193].

Mechanical drawing is another technique that has been adopted to manufacture the long hollow fibers made of silica or organic polymers, where the inner diameters of the resultant fibers are often restricted in the micrometer scale [194]. In contrast, the size of the nanotubes fabricated from layered structures are always too small and typically less than 10 nm [195,196]. Additionally, it was challenging to create sufficiently long hollow nanofibers due to the unstable connections of fibers formed during the coating and etching steps [197]. Recently, Xia et al. reported a kind of hollow fiber with controllable dimensions ranging from 20 nm to 1 mm. This hollow fiber could be easily fabricated by coaxial electrospinning two immiscible liquids, followed by selective removal of the core [197]. Such a technique provides a highly versatile method for obtaining tubular nanofibers at a large scale.

A schematic illustration of the coaxial electrospinning setup for the hollow nanofibers is presented in Figure 7A. It should be noticed that a spinneret composed of two coaxial capillaries is necessary for the development of hollow nanofibers. This spinneret was fabricated by inserting a polymer-coated silica capillary into a stainless-steel needle. In the typical procedure, heavy mineral oil and ethanol solution consisting of PVP/Ti(OiPr)_4_ are simultaneously fed through the inner and outer capillaries, respectively. During the spinning process, the jet will be subsequently stretched by electrostatic repulsions between the surface charges to generate coaxial nanofibers. Afterward, there are two choices to acquire the nanofibers with hollow interiors successfully. One is extracting the mineral oil phase with octane; the other is calcining the fibers in the air at 500 °C for 1 h to eliminate both PVP and oil phases simultaneously. It can be seen from the TEM images that the uniform tubular fibers have an inner diameter and wall thickness of 200 and 50 nm, respectively, verifying that the oil phase was incorporated as a continuous thread in each fiber during the coaxial electrospinning (Figure 7B,C).

Additionally, SEM images further confirm that the ceramic hollow fibers developed by calcination in air possess circular cross-section and relatively-smooth surfaces (Figure 7D) [198]. It is interesting to find that the feeding rate of the oil phase plays a critical role in determining the diameter of the fiber. As Xia et al. once reported, the formation of continuous hollow fibers with relatively uniform size required a feeding rate of at least 0.05 mL/h. When the feeding rate was below this value, short hollow segments were formed inside each fiber, and their sizes were not uniform. On the contrary, as the oil phase was injected faster than 0.1 mL/h, the walls of these hollow fibers would become thinner, and some bigger openings would form on the hollow fibers [198].

The inner diameter and the wall thickness of the hollow fibers could be readily varied from tens of nanometers to several hundred nanometers by modulating the coaxial-electrospinning parameters [199,200]. For example, solvents with high dielectric constants will reduce the fiber diameter [198]. The molecular weight of the polymers and the concentration of the polymer solutions are also the predominant factors that influence the fiber morphology due to the polymer entanglement effect. It was discovered that the overall diameter of the hollow fibers, as well as the wall thickness, increased when the entanglement of polymers intensified [201].

In addition to controlling the morphology of the fibers, the coaxial electrospinning technique is also capable of generating hollow nanofibers with controllable hierarchical structures and multiple functionalities. The inner and outer surfaces of tubular fibers could be independently-decorated through depositing functional molecules or nanoparticles onto the oil phase or the polymer solutions [202,203]. It is interesting to note that decoration of the surfaces of hollow nanofibers is hugely beneficial to applications in chemical sensing, where surface performance plays a vital role in determining the devices selectivity. For instance, coupling TiO_2_ with SnO_2_ or other types of metal oxides in the form of a core/shell structure could significantly improve the photoelectrical conversion efficiency of TiO_2_-based solar cells [204]. In summary, the capability to produce uniform hollow structures with tailor-made surface features will allow coaxial electrospun fibers to be particularly-customized for a variety of specific functions such as nanofluidic channels, drug delivery, gas storage, and sensing.

### 3.3. Porous Structures

The performance of the materials strongly depends on their surface topology and textural properties. As the structure of polymer fibers is switched from a solid structure to a porous one, several characteristics are also altered including increased specific surface area, large porosity, small interwoven pores, and functional versatility. These properties are particularly-favorable for fulfilling a wide range of applications, including ultra-filtration [205] absorption [206], ion-exchange [207], and as a support or carrier for reagents and catalysts [208]. Researchers have devoted a significant amount of effort toward acquiring porous structures and further promoting their practical applications. Three slightly different approaches have been summarized for generating the porous electrospun fibers with excellent mechanical strength and enhanced absorbing capacity.

One of these methods was based on the selective removal of a single component from the nanofibers, where the electrospinning solutions are possibly made of composite material [209], polymer blends [210], or organic block copolymers [211]. After leaching out one phase from the complex system, the resultant scaffolds exhibited a novel porous structure containing a dual-porosity network in the ranges of a few nanometers to a few hundred micrometers. The developed nanofibers maintained structural integrity successfully during the complete biodegradation reactions, demonstrating their enhanced potential for being utilized as engineering scaffolds. Wendorff et al. investigated the structural changes for PLA/PVP blended fibers through selectively dissolving PVP by water or removing PLA by an annealing treatment at elevated temperatures [212]. It was found that when PVP and PLA were in equal portions, the fibers would become porous and displayed a regular surface structure. If the minor component with a fraction below 50 wt% was removed, the fibers would remain compact in the structure without any visible alteration of the surface morphology. In contrast to the solid electrospun fibers, porous polymer fibers exhibit superior properties, including a low specific gravity, multiform framework, and large surface area. Therefore, it could be expected that by selective removal of one component, porous fibers have the potential to be of great interest for the preparation of functional fibers.

The second method of producing porous fibers involves the use of phase separation provided by judicious selection of the spinning parameters and the solvent types. Figure 8 shows the formation mechanism of the PMMA fibers. It was suggested that the rapid evaporation of a more volatile solvent, such as DCM, might induce a decrease in the temperature and condensation of water vapor. The polymers were separated into different phases, and the solvent-rich regions were transformed into pores on the electrospun fibers [213,214]. Compared to indirect methods to generate pores, such as selective dissolution of one component, this approach does not require any post-electrospinning treatments and is superior in terms of practicality.

In addition, it should be noted that the moisture in the environment and the solvent vapor pressure are two important factors that can strongly affect the size and density of the circular pores [215]. At low humidity and temperature, a small amount of water diffuses into the solution jet and causes a delayed solidification. Consequently, the capillary instability overcomes the viscoelastic stresses resulting in the development of beads or bead-on-string morphologies. In contrast, at increased humidity and temperature, water vapor easily condenses into droplets and attaches properly on the fiber surfaces, promoting the formation of amount of pores. Therefore, a relatively higher humidity level will cause an increase in the pore size until the pores eventually lose their uniform porosity [216].

Based on our previous work, porous fibers with 3 nm mesopores to 450 nm macropores can also be obtained by employing the following synthesis procedures, in which the pore diameters are very sensitive to the degree of fibrous shrinkage [217,218]. BG fibers were first prepared by electrospinning a transparent silica sol containing bioactive components followed by calcination at 600 °C [219]. Afterward, the corresponding nanoporous BG fibers can be precisely organized into 3D macroporous scaffolds at the macroscopic scale. The designed BG fiber membranes with hierarchal porosities have great potential in the applications of drug delivery, bone tissue engineering, and wound healing [220]. It was demonstrated that through a biomineralization reaction, porous BG fibers were tightly-integrated with a kind of doxorubicin hydrochloride to form the drug-loaded composite fibers. The drug release profiles showed that the as-synthesized fibers were acid-sensitive and drugs could be effectively-released at acidic conditions (pH 5.0), but not at neutral conditions (pH 7.4).

In addition, Kalra et al. reported the formation of the uniformly porous carbon fibers with extremely high surface area, which were fabricated by electrospinning a blend of polyacrylonitrile (PAN) and Nafion, followed by high-temperature carbonization [221]. These porous carbon nanofibers mats regarded as the free-standing electrodes for supercapacitors exhibited an ideal capacitive behavior with a large specific capacitance of greater than 200 F/g. This was attributed to the existence of a hierarchical porous structure in these carbon nanofibers, which endowed the mats a high specific surface area of up to 1600 m^2^/g and a significant fraction of mesopores (2–4 nm).

### 3.4. Multilayer Structures

Electrospinning is an efficient and straightforward method that works in a simple manner to obtain micro- and nano-scale polymer structures. Recent advances in this field have shown that electrospun biopolymer nanofiber mats could be used to create single or multi-layer, LbL assembly, and similar structures [222]. In particular, the multi-layer system containing electrospun ultra-thin fibers could significantly improve the barrier properties of biopolymers, provide high specific surface area for compound diffusion, and minimize the loss of active compounds.

The multi-layer structure is generally obtained by depositing electrospun monolayer and multi-layer on the paper substrate with two kinds of collector: plate collector and drum collector. According to the literature, multi-layer assembly of biopolymers with complementary properties behaves better in packaging materials for meeting more specific requirements of different foods [223]. For the first time, fiber-based packaging materials were prepared by electrospinning and coating with different biopolymers (poly-β-hydroxybutyrate PHB, PVA, PLA) [224]. The synthesized multilayer film was then annealed to obtain a transparent continuous electrospun film, which improved adhesion of the film to the paper substrate, enhanced the barrier property, and presented potential application prospects in the field of fiber food packaging.

Likewise, multi-layer wound dressing electrospun nanofiber mats have been created in wound treatment applications, which are more attractive than the pristine mats due to their enhanced properties [225]. A kind of new double-layer electrospun composite nanofiber mat with dual functions has been proposed recently for wound-dressing applications [226]. The first layer of the mat that consisted of PVA/chitosan/AgNPs was exposed to the environment, where AgNPs were used as a protective layer to protect against environmental microbial invasion. The second layer was in direct contact with the damaged part. It was composed of electrospun PEO or PVP nanofibers and combined with chlorhexidine as a model antibacterial (antiseptic) compound. In such layer, the involvement of chlorhexidine inhibited bacterial growth at the wound site, so as to promote the healing process.

In the fields of sensing technology, optoelectronics, etc., many nanofiber mats are also assembled by using a large voltage source in the as-spun fiber to provide higher voltage and current output than a single-pad device [227]. Figure 9 shows a robust packaging method that depends on a multilayer electrospun nano-fiber mat. As the nanofiber number increases, the piezoelectric layer integrated within the multilayer system enhances the output voltage and current. Along with the increment of device capacity, the bead array in the electrode could further increase the piezoelectric output.

During the electrospinning, PVDF-TrFE particles (70/30 Mol%) were dissolved in a mixture of acetone and dimethylamine (volume ratio of 4:6) at a concentration of 14% (*W*/*V*) to obtain a polymer solution. The bead array was transferred to a support piece coated with a silver conductive paste to form a beaded electrode as shown in Figure 9A. For the core of nanofiber mat, a pair of electrodes are assembled into a sandwich and packaged in a thin nylon-polyethylene composite film to encapsulate the entire assembly. As shown in Figure 9B, SEM images showed the morphology of primary fiber mat. The constituent fibers had a non-defective and continuous form with a nanometer-diameter of 435 ± 84 nm. For micro-morph-based electrodes, the topography was achieved using an array of beads consisting of many conductive pads.

In order to evaluate the piezoelectricity produced by our proposed structure, several different types of piezoelectric devices were prepared, including different components and packaging methods, and then were characterized by tight packaging, multilayer stacking, and microtopography-based integration. Micro-deformation was caused by integration, whereby double-sided integration could lead to significant deformation of the nanofiber mat, which was naturally consistent with the morphology of the bead array. Due to the regular arrangement of the microbeads, the core nanofiber mat was embedded in a periodic-corrugated shape with almost no voids in the filling device.

As shown in Figure 9C, the output of a typical piezoelectric system increased in proportion to the applied pressure. Also, in our device (double-side bead array (P-DB5)), the output of both voltage and current were augmented by increasing the applied pressure. When the pressure that applied to the device was up to 288 kPa, an output voltage and current of 10.4 V and 2.3 μA were generated, respectively. As predicted, the bead-embedded device showed up higher voltage and current values than the beadles’s flat device. Piezoelectric devices were used not only in energy generators but also in highly-sensitive mechanical sensors. To determine its suitability for high-sensitivity devices, the piezo device was set up and tested, using a custom measurement system equipped with a function generator, an electric vibrating screen for load cells, and a data acquisition module.

The physical sensing performance of the device was verified, as shown in Figure 9D. When light objects such as a small leaf (30 mg), rice (25 mg), and a drop of water (25 mg) were applied to the device, an identifiable peak signal corresponding to these objects was produced. On the whole, such enhanced piezoelectric device consisting of a variety of components was a high-performance generator and high precision sensor. By integrating multiple functional materials into the device, more applications could be provided for piezoelectric systems.

### 3.5. Side-by-Side Structures

Nanomaterials prepared by electrostatic spinning can be designed into two separated parts in accordance with chemical composition and function, which have received extensive attention in recent years [228]. Considering a two-compartment system, there are two relationships between components are feasible. One is external and internal (i.e., core Coaxial electrospinning or emulsion electrospinning, are other promising ways in producing the core–shell fibers to preserve the activities of proteins shell structure), and the other one is side by side, where the sides of the structure are different. Both of them can be used to develop materials with adjustability or versatility [229]. Core–shell structures produced by electrospinning, such as electrospun fibers, nanotubes, electrospray particles, and bubbles, have been extensively probed. More complex structures have also been reported, for instance, three layers of nanofibers and microparticles obtained from a triaxial electrospinning process [230], as well as multi-chamber structures manufactured from multiple fluid spinners. Unlike the core–shell structure, the latter heterojunction structure provides an opportunity to directly interact with the surrounding environment for both components, which can be advantageous for designing novel features.

From this point of view, side-by-side structures are commonly found in nature and have recently become a research hotspot for researchers, which are more attractive for the manufacture of multifunctional nano-products than core–shell structures [231]. Side-by-side electrospinning involves a complex interplay among fluid dynamics, electrodynamics, and rheology, presenting a significant challenge in controlling the movement of two fluids synchronously in a side-by-side manner under an electrical field from the spinneret to collector.

As far as we know, Gupta and Wilkes first reported the preparation of side-by-side polymer nanofibers based on a spinneret consisting of two parallel Teflon capillaries, which were made of poly (vinyl chloride)/segmented PU and poly (vinyl chloride)-PVDF [232]. After that, researches associated with side-by-side nanofibers and their corresponding applications have continuously been reported. First, structurally adjustable side-by-side fibers can be created by using a series of spinners with different port angles. Additionally, by controlling the electrospinning parameters precisely, side-by-side fibers possessed with different width and interface area can be fabricated, resulting in the volume-adjustable structure on both sides [233].

A uniform bio-based PLLA and *Bombyx mori* silk (SF) fibroin two-in-one fiber has been reported recently [234], using a side-by-side electrospinning nozzle. Such silk-based electrospun fibers with β-sheet structures exhibited a tensile strength of 16.5 ± 1.4 MPa, modulus of 205 ± 20.6 MPa, and an elongation rate at break of 53 ± 8%, where the values were very similar with those of the fibers made from a blend of SF and PLLA. It would be interesting to use such fibers to provide a new platform for designing multiple-functional materials and developing novel nanostructures, finally applying in several areas of biodegradation studies, cell culture, scaffold, and drug-release depending on the side-by-side morphology and surface chemistry of two sides.

Liu et al. have proposed a novel side-by-side microfiber membrane (UFM) consisting of PAN/PVP using electrospinning technology, which has been successfully applied to biphasic drug release [235]. Taking advantages of the self-supporting property and the differing dissolving properties, the PAN/PVP Janus UFMs could serve as a drug carrier. At the same time, two fluorescent dyes were added on both sides to study the drug release trend. Due to the different properties of the two polymers, UFM showed two-phase drug release, which could provide an adequate “loading dose”, increasing the plasma concentration of the drug to rapidly and quickly relieve the symptoms of the patient. The other one was maintaining an effective therapeutic concentration in the subsequent extended release period to avoid repeated administration.

Accurate control of the sustained release rate is important to ensure the most effective and safe pharmacokinetic characteristics of a particular disease, as well as to promote the maximum absorption of oral drugs. As shown in Figure 10, Yu et al. have reported a Teflon-coated spinneret, which could be employed to prepare a series of efficient and stable side-by-side electrospinning [236]. Taking PVP K60 and ethyl cellulose (EC) as raw materials and ketoprofen (KET) as an active ingredient, two different sides were prepared. In some cases, PVP K10 was added to the EC side of the fiber as a porogen. Electron microscopy images clearly show the generation of integrated side-by-side fiber structures, in which an amorphous distribution of KET was found. A biphasic drug encapsulated inside the fibers were released into the solution after a burst initial release. In vitro dissolution tests showed that all the fibers were capable of providing a biphasic controlled-release curve. The release rate and total release percentage can be precisely-adjusted by varying the amount of PVP K10 doped on the EC side of the fiber.

In summary, nanomedical delivery systems with highly-adjustable release profiles can be successfully prepared by the side-by-side electrospinning method, which are difficult to achieve through conventional pharmaceutical techniques. This work provides an adjustable release profile that may bring a wide range of new drugs to complement the natural biological rhythm for achieving the maximum therapeutic results.

## 4. Challenges and Future Perspectives of Electrospun Polymer Nanofibers

During the past 20 years, electrospinning has made a huge leap in the field of nanotechnology. It has proven to be a powerful technology to create a variety of functional nanostructures for different applications, as discussed throughout the review. Compared with traditional nanofiber preparation technology, electrospinning can produce fibers with high specific surface area, uniform pore size, and high porosity, which significantly improves the performance of nanofibers. In practical terms, the versatility of electrospinning has been extensively-studied and has begun to enter the industrial market. Specifically, some leading companies including DuPont, Ahlstrom, Donaldson, etc., have developed electrospinning-related products for filtration.

Furthermore, the electrospinning process is a reliable technique for designing nanofiber structures through operating conditions, such as polymer concentration, solvent choice, molecular weight, and conductivity. Meantime, various biopolymers of chitosan, cellulose, lignin, PLA, PCL, PEO, and PVA have been employed to fabricate a variety of nanostructures individually or in combination. These well-organized nanofiber structures have their broad applications, like packaging, drug delivery, filtration, fuel cells, and so forth. Compared with the conventional pharmaceutical technology, side-by-side fiber structures obtained by electrospinning have the ability to realize a novel two-phase drug release. Such sustained-release behavior can increase the plasma concentration of the drug as well as rapidly relieve the symptoms of patients through providing an effective “loading dose”.

Despite these abovementioned advantages, there are still some challenges in ahead that need to be addressed before realizing the clinical applications of electrospun mats, including the accurate and reproducible control of fiber morphology, structure, as well as uniformity. Additionally, manufacture of the electrospun scaffolds with clinically-relevant dimensions remains a challenge. In detail, despite the high adjustability and relatively-low cost, the collection speed of electrospinning is relatively slow, which raises concerns about the scale of the electrospinning process. Last but not least, considering that the lack of cell infiltration severely restricted the biomedical applications of electrospun membranes, some new technologies of reducing fiber-packing density, multilayer electrospinning, cell electrospray, and dynamic cell culture have been proposed recently to overcome the drawback. Although with these as-faced challenges, the versatility of electrospinning nanofibers combined with innovative nanostructures exhibit promising potential in many research areas.

## Figures and Tables

**Figure 1 molecules-24-00834-f001:**
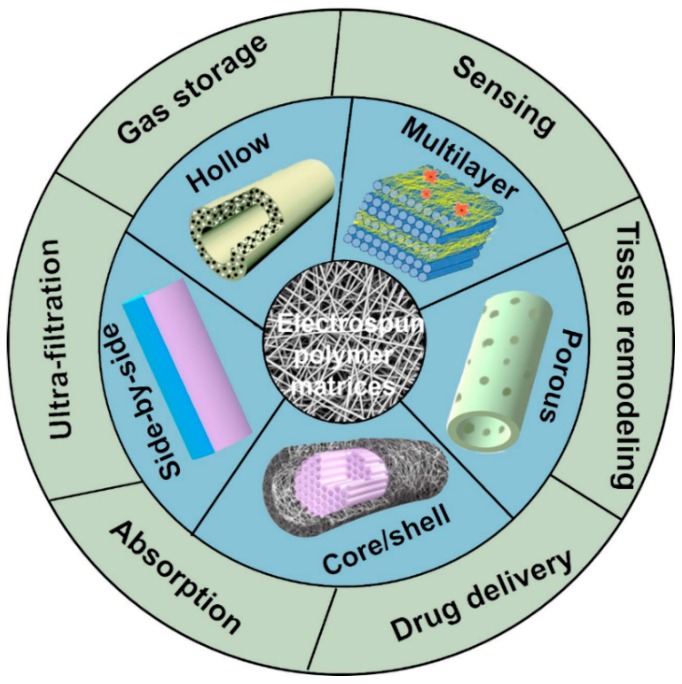
Possible structures and applications of electrospun fibers.

**Figure 2 molecules-24-00834-f002:**
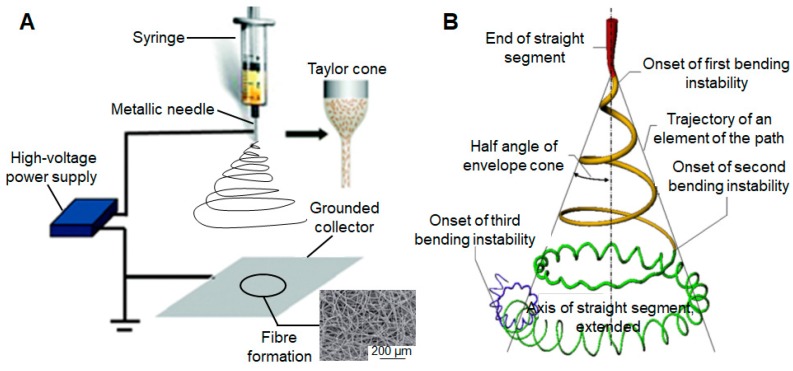
The primary electrospinning apparatus and mechanism. (**A**) Schematic illustration of the typical electrospinning set-up; reproduced from [38] with permission from the Royal Society of Chemistry, Copyright 2014; (**B**) A diagram that shows the prototypical instantaneous position of the path of an electrospinning jet that contained three successive electrical bending instabilities; reproduced from [39] with permission from Elsevier Ltd, Copyright 2008.

**Figure 3 molecules-24-00834-f003:**
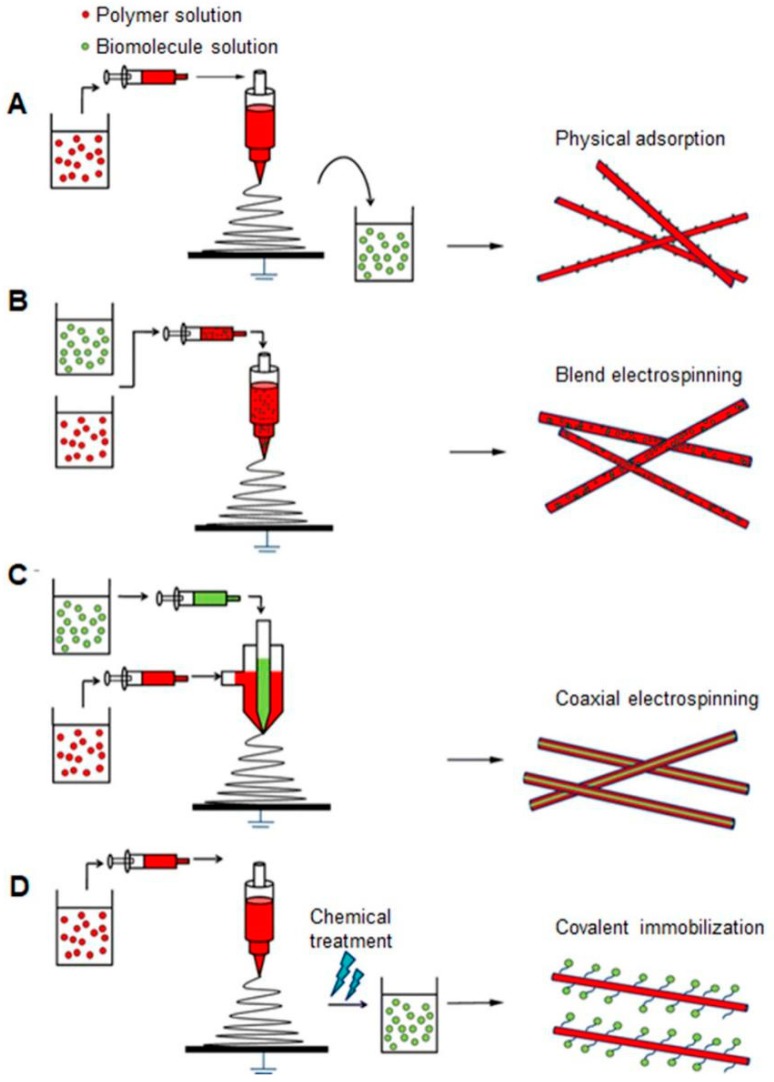
Fabrication techniques of bioactive electrospun scaffolds (**A**) physical adsorption; (**B**) blend electrospinning; (**C**) coaxial electrospinning; (**D**) covalent immobilization. Reproduced from [79] with permission from Springer Nature; Copyright 2008.

**Figure 4 molecules-24-00834-f004:**
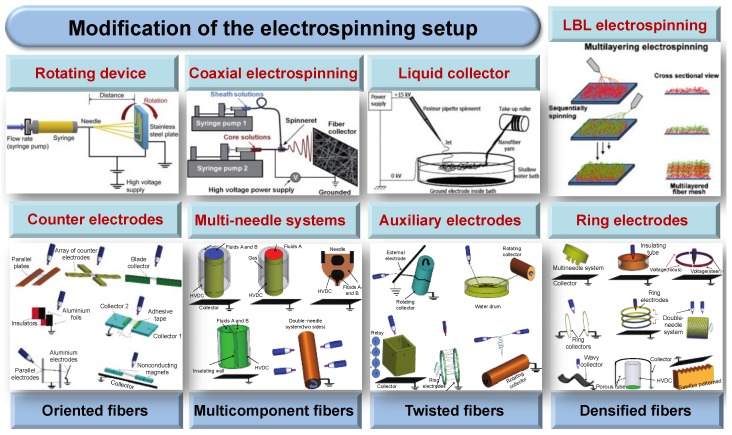
A schematic diagram of the modified electrospinning set-up. Reproduced from [145] with permission from Elsevier Ltd.; Copyright 2008. Reproduced from [146] with permission from Future Science Group; Copyright 2012. Reproduced from [140] with permission from Elsevier Ltd.; Copyright 2005. Reproduced from [129] with permission from Elsevier Ltd.; Copyright 2005. Reproduced from [147] with permission from Hindawi; Copyright 2011.

**Figure 5 molecules-24-00834-f005:**
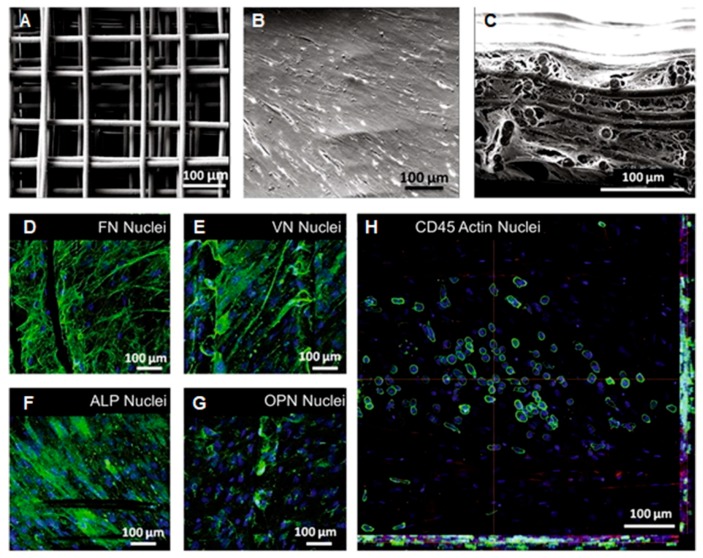
Melt electrospun scaffolds and their effects on human hematopoietic stem cells. (**A**) Melt electrospun PCL scaffolds. (**B**) Scanning electron microscopy (SEM) image of primary human osteoblasts on scaffolds in osteogenic conditions. (**C**) Cross-section profiles of scaffolds seeded with primary human osteoblasts in osteogenic conditions. (**D**–**G**) The expression of (**D**) fibronectin (FN), (**E**) vitronectin (VN), (**F**) Alkaline phosphatase (ALP), and (**G**) osteopontin (OPN) by primary human osteoblasts in osteogenic conditions. (**H**) Confocal microscopy showed that CD45^+^ cells (green) attached and migrated into primary human osteoblast-seeded scaffolds using osteogenic conditions (red); cell nuclei in blue. Reproduced from [154] with permission from Elsevier Ltd.; Copyright 2017.

**Figure 6 molecules-24-00834-f006:**
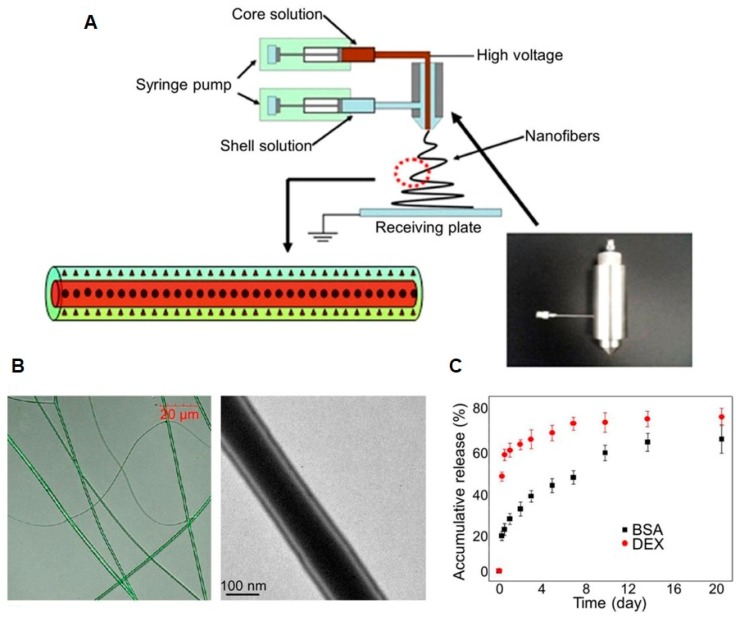
Preparation and characterization of dexamethasone (DEX) and bone morphogenetic protein-2 (BMP-2) co-loaded emulsion electrospun nanofibers and their effects on human mesenchymal stem cells (hMSCs). (**A**) Preparation of poly(L-lactide-*co*-caprolactone) (PLLACL) emulsion electrospun nanofibers with DEX in the shell and BMP-2 in the core. (**B**) Fluorescence microscopic image and transmission electron microscopy (TEM) image of the nanofibers. (**C**) Release profiles of bovine serum albumin (BSA) and DEX from the nanofibers. Reproduced from [180] with permission from Elsevier Ltd.; Copyright 2012.

**Figure 7 molecules-24-00834-f007:**
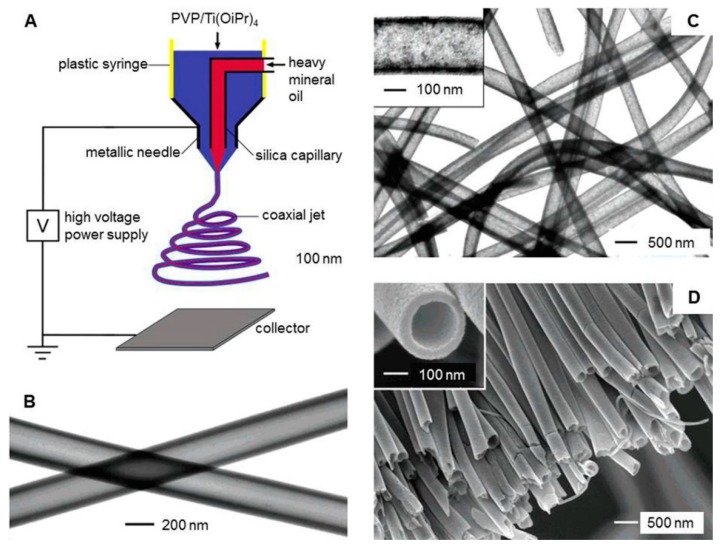
Preparation and characterization of electrospinning nanofibers with hollow structures. (**A**) Schematic illustration of the electrospinning setup for hollow structure nanofibers. (**B**) TEM image of the hollow fibers. The walls of these tubes were composed of amorphous TiO_2_ and PVP. (**C**) TEM image of TiO_2_ hollow nanofibers that were obtained by calcining the composite tubes in the air at 500 °C. (**D**) SEM image of TiO_2_ hollow fibers that were in a uniaxially aligned array. Reproduced from [198] with permission from the American Chemical Society; Copyright 2004.

**Figure 8 molecules-24-00834-f008:**
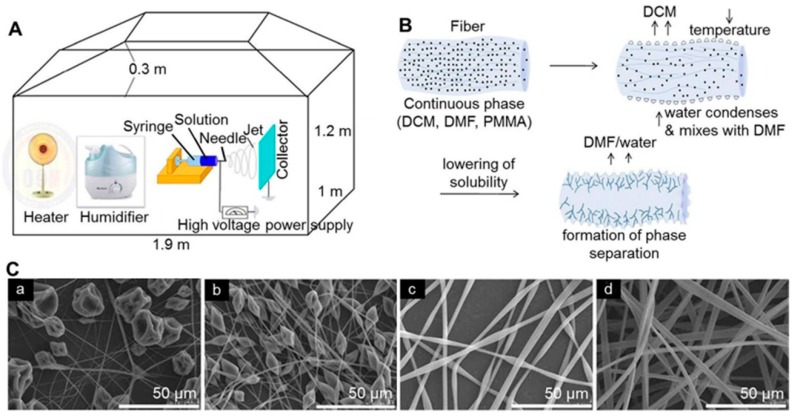
Preparation and characterization of electrospinning fibers with porous structure. (**A**) Schematic illustration of the electrospinning chamber for porous structure microfibers. (**B**) Schematic diagram showing the formation of pores during electrospinning. (**C**) SEM images of the electrospun PMMA fibers obtained at polymer concentrations of (a) 12, (b) 15, (c) 18, and (d) 21 wt% and at a humidity of 25%. Reproduced from [214] with permission from Springer Nature; Copyright 2013.

**Figure 9 molecules-24-00834-f009:**
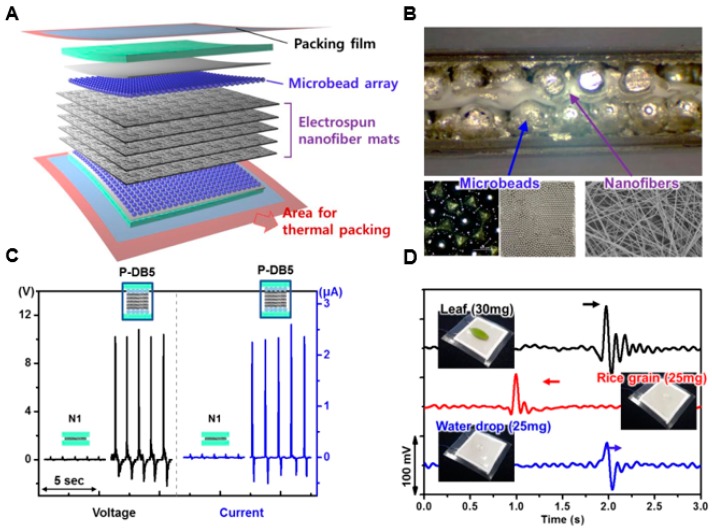
Structure and central performance characterization of multilayer electronic device. (**A**) Illustration of the overall components of the piezoelectric device with microscopic photographs of the microbead array. (**B**) Cross-sectional images of the vacuum-packed devices with P-DB5 and an SEM image of the electrospun nanofiber mat. (**C**) Relationship between the output performances of voltage and current varied by the applied pressure using the device with P-DB5. (**D**) Output signals of the drop test results using a small leaf, a grain of rice, and a water droplet. Reproduced from [227] with permission from the American Chemical Society; Copyright 2018.

**Figure 10 molecules-24-00834-f010:**
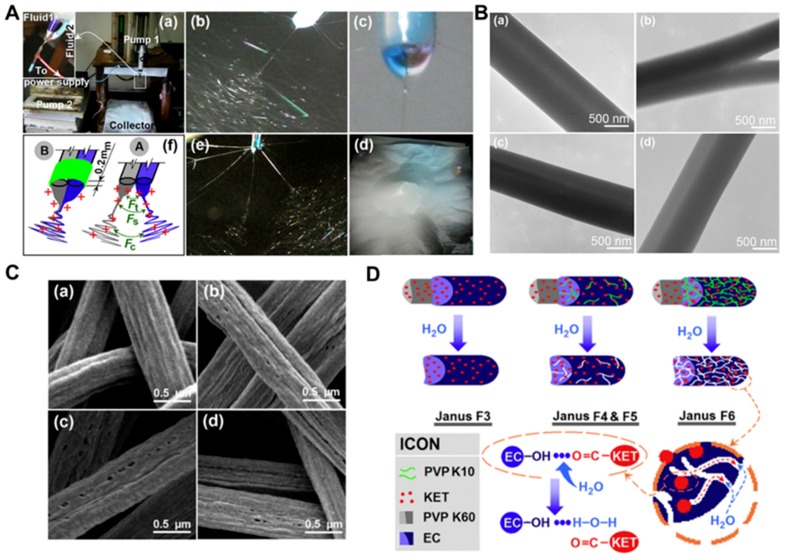
Fabrication, morphology, and mechanism of side-by-side structures. (**A**) Side-by-side electrospinning process: (a) experimental apparatus (inset: connection of side-by-side spinneret with the working fluids and power supply); (b) a photograph of a typical side-by-side electrospinning process with the Teflon-coated spinneret; (c) a Janus Taylor cone formed with the Teflon-coated spinneret; (d) fiber mat from side-by-side electrospinning with uncoated side-by-side spinneret; (e) the separation of fluids when using the uncoated spinneret; (f) an illustration of the role played by the Teflon coating: A—the separation of fluids arising from repulsive forces Ft (between the two Taylor cones), Fs (between the two straight fluid jets). (**B**) TEM images of (a) F3; (b) F4; (c) F5; (d) F6 and Fc (between the two coils); and B—the formation of an integrated Janus Taylor cone with the Teflon coating. (**C**) Field emission scanning electron microscope (FESEM) images of the fibers remaining after 24 h of dissolution and the proposed drug release mechanism. (a–d) show the remains of fibers F3–F6 respectively; (**D**) is a schematic diagram explaining the mechanism of drug release from the Janus fibers. Reproduced from [236] with permission from Elsevier Ltd.; Copyright 2016.

**Table 1 molecules-24-00834-t001:** Some fabrication methods for nanofiber mats.

Method	Advantages	Disadvantages	References
Phase separation	High porosity	Thin fibers and small pores	[5,6]
Template synthesis	Designed fiber morphology	Low porosity	[24]
Melt-blown	High efficiency, commercial	Instability, fiber diameter exceeding 1–2 μm	[8,9]
Self-assembly	A simple route to synthesize multifunctional nanofibers	Introduction of the organic solvent	[10,11]
3D printing	Controlled pore size	Low porosity	[25]
Electrospinning	Easy process and controlled fiber morphology	Small pores	[14,15,16]

**Table 2 molecules-24-00834-t002:** Composition, solvent, concentration, and functionality and applications of polymer fibers.

Composition	Solvent	Concentration	Functionality and Applications	References
Polymetylmethacrylate (PMMA)	Tetrahydrofuran (THF), acetone, chloroform	10 wt%	Superhydrophobic units for active packaging	[44]
Polyvinyl alcohol (PVA)	DI water	8–16 wt% and 1–10 wt%	Biofilters and biomembranes	[45,46]
Poly (lactic-*co*-glycolic acid) PLGA	Polysorbate 80, ethanol/ethyl acetate	4 wt%	Produced by a low-energy nano-emulsification approach, an easily scalable methodology, appropriate for the pharmaceutical industries	[47]
Polycaprolactone (PCL)	Chloroform and acetone	10% (*W*/*W*)	Show great potential for further formulation as oromucosal drug delivery systems	[48,49]
Poly (L-lactic acid) (PLLA)	*N*,*N*-dimethyl-formamide (DMF) and dichloromethane (MC)	10 wt%	Sterilize PLLA membranes for clinical applications in regenerative medicine	[50]
Gelatin	DI water	30–50% (*W*/*V*)	For tissue regeneration, the versatility of this biomaterial	[51]
Chitosan	Trifluoroacetic acid (TFA)	1–6 wt%	Tissue engineering properties and wound healing	[52,53]
Starch	Dimethyl sulfoxide (DMSO), glutaraldehyde	25 wt%	Applications in the fields of tissue engineering, pharmaceutical therapy, and medical	[54]
Collagen	TFA	42.85% (*W*/*W*)	Supports cell attachment and growth, form fibrous tissue engineering scaffolds	[55]
PLGA-curcumin	Chloroform/methanol	40 wt%/60 wt%	Delivering curcumin over a long period in a controlled manner	[56,57]
PLGA–collagen	Hexafluoroiso-propanol (HFIP)	20% (*W*/*V*)	For bioengineered skin substitutes	[58]
PCL–chitosan	HFIP and acetic acid	20:1 (*W*/*W*)	The fast degradation profile leads to rapid cellular infiltration, improved vascular remodeling, and neotissue formation without calcification or aneurysm	[59]
Poly(ε-hydroxybutyrate-*co*-ε-hydroxyvalerate) PHBV–gelatin	Tetrafluoro-ethylene (TFE)	50 wt%	Serves as a useful alternative carrier for ocular surface tissue engineering and use as an alternative substrate to amniotic membrane	[60,61]
Hydroxyapatite (HAP)–tussah silk fibroin	Ammonia, citric acid	31 wt%	Supply as scaffolds in tissue engineering and bone regeneration	[62]
Poly(lactic acid) (PLA)/PCL–cellulose nanocrystals	Acetone, DCM, toluene with phosphorus pentoxide	1wt%	Biodegradable character, use in different areas such as biomedicine or food packaging	[63]
PVA/alginate-bioglass	DI water	10 wt%	With proper biological and mechanical properties for soft/hard tissue applications	[64,65]
Polycatecholamine/CaCO_3_-collagen	HFIP, CaCl_2_ solution	8% (*W*/*V*), 10% (*W*/*W*)	Provide multifunctional scaffold properties for possible bone tissue engineering applications	[66]
PCL/(polyvinylpyrrolidone) PVP-*trans*-anethole	Chloroform: methanol	10% (*W*/*V*), 30% (*W*/*V*)	Promoting in vitro osteoblast differentiation, we can help with site-specific repair and regeneration of bone tissue	[67]
Polyurethane (PU)–dextran–estradiol	DMSO and THF	10 wt%	Post-menopausal wound dressing	[68]
PVA–PVP–HAP	DMSO	2.5, 5, 8.5, 10, and 15 wt%	Sensor, anti-static, microwave absorbing, and conductive coating	[69,70]
PLGA–tussah silk–graphene oxide	HFIP	13 wt%	Cancer treatment, therapeutic patch for drug delivery, and an excellent scaffold material for bone tissue engineering	[71]
Polyvinylidene fluoride (PVDF)–graphene oxide–silver	Acetone and DMF	2 wt%	Micro and nanoscale magnetoelectric devices, magnetic-field sensors, and energy-harvesters	[72,73]
Poly (ε-caprolactone)–cellulose acetate–dextran–tetracycline hydrochloride	DMF, THF	10 wt%	Good bioactivity, high cell attachment and proliferation, effective antibiotic activity against bacteria, for wound dressing and skin engineering applications	[74]

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
