# Peer review of "Fabrication of Electrospun Polymer Nanofibers with Diverse Morphologies"

_molecules, 2019, doi:10.3390/molecules24050834_

Reviewer 1 Report

This review is interesting, well argued and fits well with the scopes of the journal.

By focusing on the different morphologies that can be realized via electrospinning, the paper evidences the high versatility of this technique and the different application fields. Even if the topic is already reported in literature, in this review a great number of data are reported and supported by references; moreover, paragraphs structure are well defined so the paper is clear and easy to read. I only suggest introducing references in Table 2 for each system reported in order to make easier the connection with literature.

Furthermore, the conclusion are consistent with both premises and discussion.

Author Response

Response to Reviewer 1:

This review is interesting, well-argued and fits well with the scopes of the journal. By focusing on the different morphologies that can be realized via electrospinning, the paper evidences the high versatility of this technique and the different application fields. Even if the topic is already reported in literature, in this review a great number of data are reported and supported by references; moreover, paragraphs structure are well defined so the paper is clear and easy to read. I only suggest introducing references in Table 2 for each system reported in order to make easier the connection with literature. Furthermore, the conclusion is consistent with both premises and discussion.

Response: Much appreciated for your kind reminding. Following your advice, we have cited some relevant references in the revised Table 2 and listed as below:

Table 2. Composition, solvent, concentration, and functionality and applications of polymeric fibers.

Composition

Solvent

Concentration

Functionality and applications

References

Polymetylmethacrylate   (PMMA)

Tetrahydrofuran   (THF), acetone, chloroform

10 wt%

Superhydrophobic   units for active packaging

[43]

Polyvinyl   alcohol (PVA)

DI water

8–16 wt%   and 1–10 wt%

Biofilters   and biomembranes

[44,   45]

Poly (lactic-co-glycolic   acid) PLGA

Polysorbate   80, ethanol/ethyl acetate

4 wt%

Produced   by a low-energy nano-emulsification approach, an easily scalable methodology,   appropriate for the pharmaceutical industries

[46]

Polycaprolactone   (PCL)

Chloroform   and acetone

10% (W/W)

Show great   potential for further formulation as oromucosal drug delivery systems

[47,   48]

Poly   (L-lactic acid) (PLLA)

N,   N-dimethylformamide (DMF) and dichloromethane (MC)

10 wt%

Sterilize   PLLA membranes for clinical applications in regenerative medicine

[49]

Gelatin

DI water

30–50% (W/V)

For   tissue regeneration, the versatility of this biomaterial

[50]

Chitosan

Trifluoroace   acid (TFA)

1–6 wt%

Tissue   engineering properties and wound healing

[51,   52]

Starch

Dimethyl   sulfoxide (DMSO), glutaraldehyde

25 wt%

Applications   in the fields of tissue engineering, pharmaceutical therapy, and medical

[53]

Collagen

TFA

42.85% (W/W)

Supports   cell attachment and growth, form fibrous tissue engineering scaffolds

[54]

PLGA–curcumin

Chloroform/methanol

40 wt%/60 wt%

Delivering   curcumin over a long period in a controlled manner

[55,   56]

PLGA–collagen

Hexafluoroisopropanol   (HFIP)

20% (W/V)

For   bioengineered skin substitutes

[57]

PCL–chitosan

HFIP and   acetic acid

20:1 (W/W)

The   fast degradation profile leads to rapid cellular infiltration, improved   vascular remodeling, and neotissue formation without calcification or   aneurysm

[58]

Poly (ɛ-hydroxybutyrate-co-ɛ-hydroxyvalerate)   PHBV–gelatin

Tetrafluoroethylene   (TFE)

50 wt%

Serves   as a useful alternative carrier for ocular surface tissue engineering and use   as an alternative substrate to amniotic membrane

[59,   60]

Hydroxyapatite   (HAP)–tussah silk fibroin

Ammonia,   citric acid

31 wt%

Supply   as scaffolds in tissue engineering and bone regeneration

[61]

Poly(lactic   acid) (PLA)/PCL–cellulose nanocrystals

Acetone,   DCM, toluene with phosphorus pentoxide

1 wt%

Biodegradable   character, use in different areas such as biomedicine or food packaging

[62]

PVA/alginate–bioglass

DI water

10 wt%

With   proper biological and mechanical properties for soft/hard tissue applications

[63,   64]

Polycatecholamines/

CaCO3–collagen

HFIP, CaCl2   solution

8% (W/V), 10% (W/W)

Provide   multifunctional scaffold properties for possible bone tissue engineering   applications

[65]

PCL/   (polyvinylpyrrolidone) PVP–Transanethole

Chloroform:   methanol

10% (W/V), 30% (W/V)

Promoting   in vitro osteoblast differentiation, we can help with site-specific repair   and regeneration of bone tissue

[66]

Polyurethane   (PU)–dextran-estradiol

DMSO and   THF

10 wt%

Post-menopausal   wound dressing

[67]

PVA–PVP–HAP

DMSO

2.5, 5,   8.5, 10, and 15 wt%

Sensor,   anti-static, microwave absorbing, and conductive coating

[68,   69]

PLGA–tussah   silk–graphene oxide

HFIP

13 wt%

Cancer   treatment, therapeutic patch for drug delivery, and an excellent scaffold   material for bone tissue engineering

[70]

Polyvinylidene   fluoride (PVDF) –graphene oxide–silver

Acetone   and DMF

2 wt%

Micro and nanoscale magnetoelectric devices,   magnetic-field sensors, and energy-harvesters

[71,   72]

Poly (ɛ-caprolactone)–cellulose   acetate–dextran–Tetracycline hydrochloride

DMF, THF

10 wt%

Good   bioactivity, high cell attachment and proliferation, effective antibiotic   activity against bacteria, for wound dressing and skin engineering   applications

[73]

43.    Chang, H.-Y.; Chang, C.-C.; Cheng, L.-P., Preparation of hydrophobic nanofibers by electrospinning of PMMA dissolved in 2-propanol and water. MATEC Web Conf. 2019, 264, 03004.

44.    Gaaz, T. S.; Sulong, A. B.; Akhtar, M. N.; Kadhum, A. A. H.; Mohamad, A. B.; Al-Amiery, A. A., Properties and Applications of Polyvinyl Alcohol, Halloysite Nanotubes and Their Nanocomposites. Molecules 2015, 20, (12), 19884.

45.    Park, M. J.; Gonzales, R. R.; Abdel-Wahab, A.; Phuntsho, S.; Shon, H. K., Hydrophilic polyvinyl alcohol coating on hydrophobic electrospun nanofiber membrane for high performance thin film composite forward osmosis membrane. Desalination 2018, 426, 50-59.

46.    Fornaguera, C.; Dols-Perez, A.; Calderó, G.; García-Celma, M. J.; Camarasa, J.; Solans, C., PLGA nanoparticles prepared by nano-emulsion templating using low-energy methods as efficient nanocarriers for drug delivery across the blood–brain barrier. J. Controlled Release 2015, 211, 134-143.

47.    Suwantong, O., Biomedical applications of electrospun polycaprolactone fiber mats. Polym. Adv. Technol. 2016, 27, (10), 1264-1273.

48.    Potrč, T.; Baumgartner, S.; Roškar, R.; Planinšek, O.; Lavrič, Z.; Kristl, J.; Kocbek, P., Electrospun polycaprolactone nanofibers as a potential oromucosal delivery system for poorly water-soluble drugs. Eur. J. Pharm. Sci. 2015, 75, 101-113.

49.    Valente, T. A. M.; Silva, D. M.; Gomes, P. S.; Fernandes, M. H.; Santos, J. D.; Sencadas, V., Effect of Sterilization Methods on Electrospun Poly(lactic acid) (PLA) Fiber Alignment for Biomedical Applications. ACS Appl. Mater. Interfaces 2016, 8, (5), 3241-3249.

50.    Ghosh, S. K.; Adhikary, P.; Jana, S.; Biswas, A.; Sencadas, V.; Gupta, S. D.; Tudu, B.; Mandal, D., Electrospun gelatin nanofiber based self-powered bio-e-skin for health care monitoring. Nano Energy 2017, 36, 166-175.

51.    Muzzarelli, R. A. A.; El Mehtedi, M.; Bottegoni, C.; Aquili, A.; Gigante, A., Genipin-Crosslinked Chitosan Gels and Scaffolds for Tissue Engineering and Regeneration of Cartilage and Bone. Marine Drugs 2015, 13, (12), 7068.

52.    Haider, S.; Al-Masry, W.; Al-Zeghayer, Y.; Al-Hoshan, M.; Ali, F., Fabrication of Chitosan nanofibers membrane via electrospinning. 2011; Vol. 1, p 810-812.

53.    Wang, W.; Jin, X.; Zhu, Y.; Zhu, C.; Yang, J.; Wang, H.; Lin, T., Effect of vapor-phase glutaraldehyde crosslinking on electrospun starch fibers. Carbohydr. Polym. 2016, 140, 356-361.

54.    Huang, G. P.; Shanmugasundaram, S.; Masih, P.; Pandya, D.; Amara, S.; Collins, G.; Arinzeh, T. L., An investigation of common crosslinking agents on the stability of electrospun collagen scaffolds. J. Biomed. Mater. Res., Part A 2015, 103, (2), 762-771.

55.    Mohanty, C.; Sahoo, S. K., Curcumin and its topical formulations for wound healing applications. Drug Discov. Today. 2017, 22, (10), 1582-1592.

56.    Esmaili, Z.; Bayrami, S.; Dorkoosh, F. A.; Akbari Javar, H.; Seyedjafari, E.; Zargarian, S. S.; Haddadi-Asl, V., Development and characterization of electrosprayed nanoparticles for encapsulation of Curcumin. J. Biomed. Mater. Res., Part A 2018, 106, (1), 285-292.

57.    Sadeghi, A. R.; Nokhasteh, S.; Molavi, A. M.; Khorsand-Ghayeni, M.; Naderi-Meshkin, H.; Mahdizadeh, A., Surface modification of electrospun PLGA scaffold with collagen for bioengineered skin substitutes. Mater. Sci. Eng., C 2016, 66, 130-137.

58.    Fukunishi, T.; Best, C. A.; Sugiura, T.; Shoji, T.; Yi, T.; Udelsman, B.; Ohst, D.; Ong, C. S.; Zhang, H.; Shinoka, T.; Breuer, C. K.; Johnson, J.; Hibino, N., Tissue-Engineered Small Diameter Arterial Vascular Grafts from Cell-Free Nanofiber PCL/Chitosan Scaffolds in a Sheep Model. PloS one 2016, 11, (7), e0158555-e0158555.

59.    Baradaran-Rafii, A.; Biazar, E.; Heidari-Keshel, S., Cellular Response of Limbal Stem Cells on PHBV/Gelatin Nanofibrous Scaffold for Ocular Epithelial Regeneration. Int. J. Polym. Mater. Polym. Biomater. 2015, 64, (17), 879-887..

60.    Choi, M.-O.; Kim, Y.-J., Effect of poly(3-hydroxybutyrate-co-3-hydroxyvalerate)/gelatin ratios on the characteristics of biomimetic composite nanofibrous scaffolds. Colloid. Polym. Sci. 2018, 296, (5), 917-926.

61.    Shao, W.; He, J.; Sang, F.; Ding, B.; Chen, L.; Cui, S.; Li, K.; Han, Q.; Tan, W., Coaxial electrospun aligned tussah silk fibroin nanostructured fiber scaffolds embedded with hydroxyapatite–tussah silk fibroin nanoparticles for bone tissue engineering. Mater. Sci. Eng., C 2016, 58, 342-351.

62.    Sessini, V.; Navarro-Baena, I.; Arrieta, M. P.; Dominici, F.; López, D.; Torre, L.; Kenny, J. M.; Dubois, P.; Raquez, J.-M.; Peponi, L., Effect of the addition of polyester-grafted-cellulose nanocrystals on the shape memory properties of biodegradable PLA/PCL nanocomposites. Polym. Degrad. Stab. 2018, 152, 126-138.

63.    Saberi, A.; Rafienia, M.; Poorazizi, E., A novel fabrication of PVA/Alginate-Bioglass electrospun for biomedical engineering application. Nanomed J. 2017, 4, (3), 152-163.

64.    İspirli Doğaç, Y.; Deveci, İ.; Mercimek, B.; Teke, M., A comparative study for lipase immobilization onto alginate based composite electrospun nanofibers with effective and enhanced stability. Int. J. Biol. Macromol. 2017, 96, 302-311.

65.    Dhand, C.; Barathi, V. A.; Ong, S. T.; Venkatesh, M.; Harini, S.; Dwivedi, N.; Goh, E. T. L.; Nandhakumar, M.; Venugopal, J. R.; Diaz, S. M.; Fazil, M. H. U. T.; Loh, X. J.; Ping, L. S.; Beuerman, R. W.; Verma, N. K.; Ramakrishna, S.; Lakshminarayanan, R., Latent Oxidative Polymerization of Catecholamines as Potential Cross-linkers for Biocompatible and Multifunctional Biopolymer Scaffolds. ACS Appl. Mater. Interfaces 2016, 8, (47), 32266-32281.

66.    PranavKumar Shadamarshan, R.; Balaji, H.; Rao, H. S.; Balagangadharan, K.; Viji Chandran, S.; Selvamurugan, N., Fabrication of PCL/PVP Electrospun Fibers loaded with Trans-anethole for Bone Regeneration in vitro. Colloids Surf., B 2018, 171, 698-706.

67.    Unnithan, A. R.; Sasikala, A. R. K.; Murugesan, P.; Gurusamy, M.; Wu, D.; Park, C. H.; Kim, C. S., Electrospun polyurethane-dextran nanofiber mats loaded with Estradiol for post-menopausal wound dressing. Int. J. Biol. Macromol. 2015, 77, 1-8.

68.    Chaudhuri, B.; Mondal, B.; Ray, S. K.; Sarkar, S. C., A novel biocompatible conducting polyvinyl alcohol (PVA)-polyvinylpyrrolidone (PVP)-hydroxyapatite (HAP) composite scaffolds for probable biological application. Colloids Surf., B 2016, 143, 71-80.

69.    Kim, G.-M.; Simon, P.; Kim, J.-S., Electrospun PVA/HAp nanocomposite nanofibers: biomimetics of mineralized hard tissues at a lower level of complexity. Bioinsp. Biomim. 2008, 3, (4), 046003.

70.    Shao, W.; He, J.; Sang, F.; Wang, Q.; Chen, L.; Cui, S.; Ding, B., Enhanced bone formation in electrospun poly(l-lactic-co-glycolic acid)–tussah silk fibroin ultrafine nanofiber scaffolds incorporated with graphene oxide. Mater. Sci. Eng., C 2016, 62, 823-834.

71.    Hong, B.; Jung, H.; Byun, H., Preparation of Polyvinylidene Fluoride Nanofiber Membrane and Its Antibacterial Characteristics with Nanosilver or Graphene Oxide. J. Nanosci. Nanotechnol. 2013, 13, (9), 6269-6274.

72.    Liu, C.; Shen, J.; Liao, C. Z.; Yeung, K. W. K.; Tjong, S. C., Novel electrospun polyvinylidene fluoride-graphene oxide-silver nanocomposite membranes with protein and bacterial antifouling characteristics. Express Polym Lett. 2018, 12, 365.

73.    Liao, N.; Unnithan, A. R.; Joshi, M. K.; Tiwari, A. P.; Hong, S. T.; Park, C.-H.; Kim, C. S., Electrospun bioactive poly (ɛ-caprolactone)–cellulose acetate–dextran antibacterial composite mats for wound dressing applications. Colloids Surf., A 2015, 469, 194-201.

Reviewer 2 Report

The paper is an interesting review showing several approaches that can be used to obtain different morphologies using electrospinning. Only some minor English double check is required

Author Response

Response to Reviewer 2:

The paper is an interesting review showing several approaches that can be used to obtain different morphologies using electrospinning. Only some minor English double check is required.

Response: Thanks for your thoughtful comment. According to your advice, we have checked the whole manuscript carefully and some language mistakes have been corrected in the revised manuscript as below:

1: “Micro/nanofiber mats have been a subject of intensive research due to the high specific surface area and the interconnected porous structure.”

2: “With the ability to fabricate nanostructures from a wide variety of raw materials, including natural and synthetic polymer composites (both organic and inorganic), electrospinning is attracting more and more scientists for the highly-efficient preparation of various nanostructures.”

3: “Unfortunately, even though with the widespread use, the understanding of the electrospinning is still very limited.”

4: “In the review, we first introduced the basic knowledge of electrostatic spinning, including device, composition, mechanism, and methods comprehensively.”

5: “The basic electrospinning apparatus consists of four major components: a high-voltage source which creates an electrical field between a positively-charged syringe needle and a grounded collector, a metallic needle where the charged solution is forced to stretch under the electrostatic forces, a syringe pump, and a grounded target to deposit the resultant fibers.”

6: “These electrospun membranes composed of copolymers, blends, or organic fillers always exhibit enhanced mechanical behavior, barrier property, and thermal stability.”

7: “A comprehensive summary of electrospun polymers, the solvents that have been used, the polymer concentrations in different solvents, and the perspective applications of corresponding nanofibers is listed in Table 2.”

8: “In order to avoid this phenomenon, researchers investigate why the burst release occurs and how to achieve a constant drug release profile.”

9: “The easiest way is to dip the scaffolds into an aqueous phase containing biomolecules, where the loaded biomolecules can attach to the scaffolds via electrostatic forces.”

10: “In the second situation, biomolecules are mixed with the polymer solution.”

11: “More importantly, the core/shell nanofibers have been reported to provide typical biphasic drug release profiles consisting of an immediate and sustained release. The amount of drug released in the first phase is tailored by adjusting the shell flowing rate, and the remaining drug released in the second phase is controlled by a typical diffusion mechanism.

12: “Through increasing the strength of the electrostatic field, the surface liquid gets electrically-charged and shear stresses are produced by the repulsion between these charges.”

13: “Natural bone is a complex biomineralized system with excellent mechanical stability, highly-densified mineralization matrix, as well as a complex hierarchical structure.

14: “Layer-by-layer (LbL) electrospinning is a good candidate for acquiring hierarchical-structured scaffolds because each polymer solution will be electrospun to form its individual layer, and subsequently deposited on the same target metallic collector in the form of a nonwoven fabric.

15: “One novel configuration composed of a multineedle system and an electrically-charged cylindrical electrode is designed to increase the deposition density of the fiber mat.

16: “This innovative approach has been employed to fabricate a vascular scaffold that dual-loaded vascular endothelial growth factor and platelet-derived growth factor. This multilayered fibrous scaffold has been demonstrated to benefit blood vessel reconstruction, facilitate endothelialization by the dual release of the growth factors, and inhibit hyperproliferation of vascular smooth muscle cells.

17: “Twisted nanofibers have been extensively exploited in the field of tissue engineering nowadays because they can readily emulate several natural materials, such as collagen fibrils and double-stranded DNA for use in clinical applications.

18: “In most cases, a solid collector is generally-applied to deposit the electrospun fibers.

19: “This was done to neutralize the free charges available on surface of the fibers.

20: “In conclusion, twisted nanofibers fabricated directly using a modified electrospinning system have great potential to be utilized in applications as artificial muscles and actuators.”

21: “Additionally, it is very difficult to fabricate melt-spun fibers with nanometer diameters and the polymers must be kept at elevated temperatures above the melting point.

22: “The melt electrospinning membrane has been widely-used as scaffolds for engineering different types of tissue.

23: “The cardiac progenitor cells were able to align more efficiently along the preferential direction of the melt electrospun pHMGCL fibers compared to commonly-used electrospun scaffolds, hereby potentially enhancing their therapeutic potential in cardiac tissue engineering.

24: “More importantly, a significantly-enhanced deposition of endosteal proteins and osteogenic markers could be observed when the tissue-engineering scaffold was combined with calcium phosphate coating under osteogenic conditions.

25: “Additionally, previous studies have shown that electrospun matrices comprised of nanofibers have extremely high specific surface area to interact with cells [135, 136], making them ideal for cell attachment and proliferation. Accordingly, an attractive feature of electrospinning is its capacity to adjust fiber size in the nanometer and submicron range, which closely resembles the size of extracellular structures.”

26: “In addition, depending on the entanglement of these micro/nanofibers, the electrospun membranes possess a highly-porous 3D network with excellent pore interconnection.”

27: “As a result, some performance differences are presented among the chain orientation of the electrospun fibers, especially the thermal behavior and physical-mechanical properties. According to Pedicini et al., in contrast to the resultant products obtained by solution casting or other conventional processes, electrospun PU fibers exhibit a distinctly-different stress-strain response curve in the uniaxial tensile test.

28: “However, for their potential to be fully-realized and to achieve advanced performances, further design of various fibrous assemblies and morphological structures are necessary.

29: “Such composite fibers normally have a hydrophobic polymer sheath and a hydrophilic core, which would be fabricated into bioactive tissue-engineering scaffolds [155]. Confocal laser scanning microscope (CLSM) images of the resultant nanofibers which consist of green core and colorless sheath show that the boundary between them is quite sharp.”

30: “The emulsion droplets were stretched into an elliptical shape along the fiber direction, which might be caused by the relatively-rapid elongation and the quick evaporation of the solvents during the electrospinning.

31: “Such technique provides a highly versatile method for obtaining tubular nanofibers at a large scale.

32: “It can be seen from the TEM images that the uniform tubular fibers have an inner diameter and wall thickness of 200 and 50 nm, respectively, verifying that the oil phase was incorporated as a continuous thread in each fiber during the coaxial electrospinning.

33: “Additionally, SEM images further confirm that the ceramic hollow fibers developed by calcination in air possess circular cross-section and relatively-smooth surfaces (Figure 6D).”

34: “When the feeding rate was below this value, short hollow segments were formed inside each fiber and their sizes were not uniform.

35: “The inner and outer surfaces of tubular fibers could be independently-decorated through depositing functional molecules or nanoparticles onto the oil phase or the polymer solutions.

36: “In summary, the capability to produce uniform hollow structures with tailor-made surface features will allow coaxial electrospun fibers to be particularly-customized for a variety of specific functions such as nanofluidic channels, drug delivery, gas storage, and sensing.”

37: “These properties are particularly-favorable for fulfilling a wide range of applications, including ultra-filtration [174], absorption [175], ion-exchange [176], and as a support or carrier for reagents and catalysts.”

38: “It was demonstrated that through a biomineralization reaction, porous BG fibers were tightly-integrated with a kind of doxorubicin hydrochloride to form the drug-loaded composite fibers. The drug release profiles showed that the as-synthesized fibers were acid-sensitive and drugs could be effectively-released at acidic conditions (pH 5.0), but not at neutral conditions (pH 7.4).

39: “In the fields of sensing technology, optoelectronics, etc., multiple nanofiber mats are also assembled by using a largevoltage source in the as-spun fiber to provide higher voltage and current output than a single-pad device.

40: “For the core of nanofiber mat, a pair of electrodes are assembled into a sandwich and packaged in a thin nylon-polyethylene composite film to encapsulate the entire assembly.

41: “Unlike the coreshell structure, the latter heterojunction structure provides an opportunity to directly interact with the surrounding environment for both components, which can be advantageous for designing novel features.

42: “Thereafter, researches associated with side-by-side nanofibers and their corresponding applications have continuously been reported.

43: “Such silk-based electrospun fibers with β-sheet structures exhibited a tensile strength of 16.5 ± 1.4 MPa, modulus of 205 ± 20.6 MPa, and an elongation rate at break of 53 ± 8%, where the values were very similar with those of the fibers made from a blend of SF and PLLA.

44: “In summary, nanomedical delivery systems with highly-adjustable release profiles can be successfully prepared by the side-by-side electrospinning method, which are difficult to achieve through the conventional pharmaceutical techniques.

45: “In practical terms, the versatility of electrospinning has been extensively-studied and has begun to enter the industrial market.

46: “Such sustained-release behavior can increase the plasma concentration of the drug as well as rapidly relieve the patients' symptoms through providing an effective "loading dose".

47: “Although with these as-faced challenges, the versatility of electrospinning nanofibers combined with innovative nanostructures exhibit promising potential in many research areas.

48: “In such review, more emphasis is put on the construction of polymer nanofiber structures and their potential applications.”

49: “Coaxial electrospinning or emulsion electrospinning, are other promising ways in producing the coreshell fibers to preserve the activities of proteins.”

50: “Taken together, these well-organized polymer nanofibers can be of great interest for biomedicine, nutrition, bioengineering, pharmaceutics, and healthcare applications.”